# Crosstalk between SHH and FGFR Signaling Pathways Controls Tissue Invasion in Medulloblastoma

**DOI:** 10.3390/cancers11121985

**Published:** 2019-12-10

**Authors:** Anuja Neve, Jessica Migliavacca, Charles Capdeville, Marc Thomas Schönholzer, Alexandre Gries, Min Ma, Karthiga Santhana Kumar, Michael Grotzer, Martin Baumgartner

**Affiliations:** 1Department of Oncology, University Children’s Hospital Zürich, CH-8032 Zürich, Switzerland; anujaneve@googlemail.com (A.N.); Jessica.Migliavacca@kispi.uzh.ch (J.M.); Charles.Capdeville@kispi.uzh.ch (C.C.); marcthomas.schoenholzer@bluewin.ch (M.T.S.); Alexandre.Gries@kispi.uzh.ch (A.G.); Karthiga.Kumar@kispi.uzh.ch (K.S.K.); Michael.Grotzer@kispi.uzh.ch (M.G.); 2Faculty of Biology and Medicine, University of Lausanne, Biochemistry, CH-1066 Epalinges, Switzerland; Min.Ma@unil.ch

**Keywords:** medulloblastoma, FGFR, Sonic Hedgehog signaling, organotypic culture, cell invasion, signal crosstalk, MAP kinase signaling

## Abstract

In the Sonic Hedgehog (SHH) subgroup of medulloblastoma (MB), tumor initiation and progression are in part driven by smoothened (SMO) and fibroblast growth factor (FGF)-receptor (FGFR) signaling, respectively. We investigated the impact of the SMO-FGFR crosstalk on tumor growth and invasiveness in MB. We found that FGFR signaling represses GLI1 expression downstream of activated SMO in the SHH MB line DAOY and induces *MKI67*, *HES1*, and *BMI1* in DAOY and in the group 3 MB line HD-MBO3. FGFR repression of GLI1 does not affect proliferation or viability, whereas inhibition of FGFR is necessary to release SMO-driven invasiveness. Conversely, SMO activation represses FGFR-driven sustained activation of nuclear ERK. Parallel activation of FGFR and SMO in ex vivo tumor cell-cerebellum slice co-cultures reduced invasion of tumor cells without affecting proliferation. In contrast, treatment of the cells with the SMO antagonist Sonidegib (LDE225) blocked invasion and proliferation in cerebellar slices. Thus, sustained, low-level SMO activation is necessary for proliferation and tissue invasion, whereas acute, pronounced activation of SMO can repress FGFR-driven invasiveness. This suggests that the tumor cell response is dependent on the relative local abundance of the two factors and indicates a paradigm of microenvironmental control of invasion in SHH MB through mutual control of SHH and FGFR signaling.

## 1. Introduction

The crosstalk between signaling pathways in eukaryotic cells controls cellular functions involved in proliferation, differentiation, viability, and motile behavior. Although the cause of cancer is usually a genetic or an epigenetic alteration, the aggressiveness of the disease is ultimately determined by the interplay of the cell with its environment, resulting in abundant signal crosstalk in the cancer cells [1]. Medulloblastoma (MB) is the most common malignant pediatric brain tumor. It is divided into four subgroups with a total of twelve molecularly distinct subtypes [2]. In Sonic Hedgehog (SHH) MB, the crosstalk between basic fibroblast growth factor (bFGF/FGF2) activated fibroblast growth factor (FGF)-receptor (FGFR) and tumor growth factor receptor beta (TGF-β) signaling determines the migratory and the invasive capabilities of the tumor cells [3]. In granule cell precursors (GCPs), the putative cells of origin of SHH MB, bFGF abolishes the potent proliferative response of SHH and accelerates differentiation of GCPs [4]. Moreover, bFGF pre-treatment of SHH MB tumor cells derived from a Ptch+/− mouse or direct injection of high concentrations of bFGF into Ptch+/− MB tumors that were orthotopically implanted in recipient mice prevented tumor growth or caused regression, respectively [5]. This raises the question of a dichotomous pro-invasive/anti-proliferative function of bFGF in MB tumors with activated SHH signaling that may impact on the growth and the dissemination of the tumor cells.

Hedgehog (Hh) signaling is an evolutionary conserved signaling pathway. It regulates a multitude of cellular processes during development, embryonic patterning, organ morphogenesis, and growth control by regulating cell proliferation, differentiation, and migration [6]. The core components of Hh signaling are Patched1 (PTCH1), a twelve transmembrane receptor that binds Hh ligands including SHH, and smoothened (SMO), a seven-transmembrane receptor relieved from PTCH1 repression after Hh ligand binding to PTCH1. This activation of SMO causes the processing of the glioma-associated oncogene transcription factors (GLI [7]) GLI2/3 from gene repressing to inducing activity and results in the expression of the constitutive activator GLI1 [8]. In addition to the repression of SMO by PTCH1, Hh signaling is regulated by other mechanisms, including the suppressor of fused (SUFU)-mediated cytoplasmic retention and degradation of GLI and the phosphorylation of GLI1, 2, 3 by protein kinases or their acetylation, ubiquitination, or sumoylation [6]. In SHH-driven MB, the induction of GLI is repressed by the phosphorylation of GLI1 by MEKK2 and MEKK3 downstream of FGFR signaling [9]. In contrast, an earlier study in fibroblasts found that bFGF activation of FGFR could induce GLI1 transcriptional activity, and it was suggested that mitogen-activated protein (MAP) kinase and SHH signaling synergize to drive proliferation [10].

In the developing central nervous system (CNS), Hh signaling exerts pleiotropic activity and drives proliferation, specification, and axonal targeting. In the adult CNS, Hh signaling modulates self-renewal and specification of neural stem cells [11]. bFGF also tunes diverse functions in the developing and the adult brain [12,13]. It is secreted by the choroid plexus, the microglia, and the astrocytes and is concentrated on laminin-containing fractone structures in the brain [14]. bFGF accumulates in MB tumor tissue in a pattern reminiscent of tumor-infiltrating cells, and it is a potent promoter of cell dissemination by signaling via an FRS2-dependent cascade [3]. The conflicting results on the role of bFGF in MB growth and invasiveness and its regulatory impact on Hh signaling highlight the unclear functional significance of the FGFR-SHH crosstalk in MB. Moreover, whether SMO activation alters FGFR function has not been addressed. In this study, we therefore test whether combined activation of FGFR and SMO affects proliferation and invasiveness in MB cells. 

## 2. Results

### 2.1. bFGF-Induced FGFR Activation Represses SAG-Induced GLI1 Expression

SHH signaling in MB can be repressed by bFGF-induced FGFR activation [9]. To explore the FGFR-SMO crosstalk in the context of cell migration control, we used the SHH MB cell line DAOY and determined whether SMO activation could also affect FGFR signaling. In these cells and also in a mouse model of gr3 MB, FGFR activity promotes cell invasion and tumor progression [3]. In DAOY cells, parallel stimulation with both secreted SHH and bFGF represses GLI1 transcriptional activity through direct phosphorylation of GLI1 by MEKK2 and MEKK3 [9]. We first tested whether SMO agonist (SAG) also leads to SHH pathway activation in DAOY MB cells and whether GLI1 expression is altered in response to parallel FGFR activation by bFGF at a concentration that promotes invasiveness. We found that SAG treatment of DAOY cells induced *GLI 1* mRNA (Figure 1A) and GLI protein expression (Figure 1B). Co-stimulation of the cells with bFGF reduced SAG-induced *GLI1* transcription and protein expression (Figure 1A,B). Treatment of the cells with the pan-FGFR inhibitor BGJ398 rescued *GLI1* expression (Figure 1C) and GLI protein levels (Figure 1D) in the presence of SAG-bFGF co-stimulation. GLI1 was not detectable in the gr3 line HD-MBO3. In combination with SAG, BGJ398 treatment also caused a dramatic increase in *GLI1* expression, whereas BGJ398 treatment alone only moderately increased *GLI1* expression (Figure 1C) but not GLI1 protein levels. This indicates that the induction of GLI1 by BGJ398 treatment both at mRNA and protein levels is effective only when FGFRs and SMO are activated. 

Kinase inhibitors of extracellular-signal regulated kinase (ERK), phosphatidylinositol 3’kinase (PI3-K), or protein kinase C (PKC) did not rescue *GLI1* expression (Figure 1D). Thus, none of these putative effectors of FGFR alone are involved in GLI1 repression. Interestingly, epidermal growth factor (EGF) stimulation for 24 h also repressed basal and SAG-induced GLI1 (Figure 1E). Thus, receptor tyrosine kinase (RTK)-dependent repression of GLI1 is not specific for bFGF. These findings show that the activation of SMO promotes *GLI1* transcription and leads to GLI1 expression in DAOY cells. Parallel activation of FGFR signaling represses GLI1 expression both at the transcriptional and the protein level (Figure 1F). Furthermore, pharmacological repression of FRGR with BGJ398 in the presence of active SMO causes a very pronounced induction of *GLI*.

### 2.2. SMO Activation by SAG Does Not Repress bFGF-Induced Collagen Invasion

Since bFGF is a strong promoter of invasion in DAOY cells [15], we tested whether SAG treatment would affect bFGF-induced collagen I invasion. We performed spheroid invasion assays in the absence of stimulation and with SAG or bFGF alone or with a combination of both. SAG treatment did not promote collagen I invasion (Figure 2A), whereas bFGF stimulation caused a robust increase in the average distance of invasion. Co-stimulation with SAG did not significantly alter bFGF induced invasion, suggesting that there is no inhibitory crosstalk between SHH and FGFR signaling with respect to invasion control. To exclude the possibility that FGFR-mediated repression of SHH signaling (Figure 1C) impedes SAG-induced collagen I invasion, we treated bFGF-stimulated cells with BGJ398 and SAG in parallel. BGJ398 treatment completely abrogated bFGF-induced invasion (Figure 2B). Co-treatment with bFGF, SAG, and BGJ398 caused a significant increase in collagen I invasion compared to bFGF plus BGJ398 treatment alone. Although BGJ398-only treatment caused some increase in invasion, its impact on invasion was only significant on the distance of invasion (Figure 1C) as well as on the cumulated distance of invasion (Figure 1D) when SAG was also present. This indicates that SMO activation can moderately increase the invasion capabilities of the cells in vitro when FGFR signaling is repressed.

### 2.3. SMO Activity Is Necessary for Proliferation without Affecting bFGF-Dependent Gene Expression

bFGF and FGFR1-4 signal via fibroblast growth factor receptor substrate 2 (FRS2)-dependent Ras/mitogen-activated kinase (RAS/MAPK) and phosphatidylinositol 3’kinase (PI3K)/ protein kinase B (PKB) and FRS2-independent phospholipase C-gamma (PLC-γ), janus kinase-signal transducer and activator of transcription (JAK-STAT) pathways that contribute to proliferation and viability [16]. To test whether SMO activation by SAG affects FGFR signaling towards proliferation and viability, we counted the cells grown in low (1%) serum without or with bFGF or SAG or a combination of both (Figure 3A). Neither bFGF nor SAG caused a marked change in cell number, whereas the inhibition of FGFRs with BGJ398 or of SMO with LDE225 significantly reduced proliferation. To compare treatment effects in cell lines of SHH (DAOY) and group 3 (gr 3, HD-MBO3 [17]) MB, we used CellTrace Violet dye intensity as a readout. In-cell dye fluorescence was increasingly diluted with each additional generation number and thus servd as a measurand for proliferation (Appendix A). Dye dilution was comparable under control conditions in DAOY and HD-MBO3, and blockade of proliferation with mitomycin C caused a comparable dye retention per cell in both cell lines (Appendix A). LDE225 treatment completely repressed proliferation in a subset of cells between 24 and 72 h (right peak in the histogram in Figure 3B, X/Y plot of time resolved mean fluorescence in Figure 3C). In DAOY cells, BGJ398 also reduced proliferation, although this effect became evident only at 72 h. In HD-MBO3 cells, both LDE225 and BGJ398 caused a similar reduction in cell proliferation compared to control, whereas bFGF caused a moderate increase in proliferation. In contrast to DAOY cells, there was no indication of a bi-phasic fluorescence intensity distribution in HD-MBO3 after LDE225 treatment. Thus, a subset of DAOY cells is highly susceptible to SMO inhibition, and HD-MBO3 cell proliferation depends in part on active FGFR signaling. 

We next determined whether FGFR and SHH pathway modulation alters the expression of *GLI1*, *HES1*, *MKI67*, and *BMI1*, a small subset of genes involved in proliferation and differentiation control. We found that SAG treatment caused a significant increase in *GLI1* expression in DAOY cells without affecting the expression of the other genes. bFGF repressed SAG-induced *GLI1* and caused significant increases in *HES1*, *MKI67*, and *BMI1* in both cell lines (Figure 3D,E). Expressions of *GLI1* and *HES1* were also significantly increased in primary SHH MB compared to the other three subgroups (Appendix A). None of the three bFGF-induced genes were repressed by parallel SAG stimulation. Consistent with the proliferation data, BGJ398 treatment reduced *MKI67* expression in HD-MBO3 cells, an effect that was not rescued by bFGF or SAG treatment (Figure 3E). These data confirm the repressive activity of FGFR activation on *GLI1* expression and reveal *HES1* as a novel target gene of FGFR signaling.

### 2.4. SMO Activation Represses Nuclear ERK Activity after bFGF Stimulation

FGFR activation causes the induction of the MAP kinase signaling pathway and an increase in the phosphorylation of ERK on residues Thr202/Tyr204 (Figure 1B and Figure 4A). To test whether parallel SMO activation could influence the extent of ERK activation in response to growth factor treatment, we compared the level of phosphorylated ERK (pERK). We observed no reduction of pERK in cells co-stimulated for ten minutes with bFGF and SAG compared to cells stimulated with bFGF only (Figure 4A). We also observed no differences in phospho-ERK when SAG stimulation was combined with hepatocyte growth factor (HGF) or EGF. We additionally explored the levels of phosphorylated ERK in three different cellular fractions [cytosolic: (C), membranes: (M) and nucleus/cytoskeleton: (N)], encompassing the totality of all isolated proteins after 10 and 90 min of stimulation with bFGF, SAG, or the combination of bFGF plus SAG. We repeated the experiment three times and observed some variations in pERK distribution. A representative blot (90 min stimulation) is shown in Figure 4B and the corresponding quantification of pERK in Figure 4C. However, statistical analysis revealed no significant difference when bFGF-alone treatment was compared to bFGF plus SAG. We furthermore compared the cumulated pERK levels from all fractions, confirming that bFGF-induced pERK is not influenced by parallel SMO activation (Figure 4D). As we could not confirm the expected increase in nuclear pERK after bFGF stimulation by immunoblot (IB), we used a synthetic kinase activation relocation sensor (SKARS) for ERK [18,19], which is designed to measure the nuclear translocation of activated ERK in real-time. We found that bFGF (Figure 4E, left) caused a dose-dependent and persistent increase in sensor translocation. Co-stimulation of the cells with 100 ng/mL bFGF plus 100 nM SAG markedly reduced bFGF-induced sensor translocation compared to bFGF stimulation alone (Figure 4E, right). This suggested that one immediate, transcription-independent effect of SHH pathway activation is the repression of bFGF-induced nuclear ERK activity (Figure 4F).

### 2.5. Combined, Exogenous Activation of SHH and FGFR Signaling Prevents Tissue Invasion Ex vivo

To evaluate the impact of FGFR and SHH signaling on tumor cell growth and tissue invasion, we performed organotypic cerebellum slice-tumor cell co-culture [18] experiments in the absence and the presence of exogenously added bFGF, SAG, or with a combination of both. DAOY cells were implanted as spheroids and treated for five days after implantation with the different growth factors and drugs. Under control conditions, the tumor cells expanded from a spheroid into the surrounding brain tissue, which resulted in a tumor cell mass of irregular shape and low circularity (Figure 5A and Figure 6A, Appendix A). Treatment of the co-culture with LDE225 prevented the invasion of the surrounding tissue (Figure 5A and Figure 6B), resulting in a spherical, well-circumscribed tumor cell mass with significantly increased circularity (Figure 5A, Appendix A). Conversely, activation of SHH signaling with SAG increased invasion, caused reduced circularity (Figure 5A), and single cells invading deep in the tissue slice were observed (Appendix A). Co-stimulation of SAG-treated cells with bFGF did not further increase invasion; it rather caused a phenotype similar to the LDE225-treated condition characterized by reduced invasion and increased circularity of the tumor cell mass (Figure 5A and Figure 6C). We also determined the impact of the treatments on proliferation by performing EdU staining followed by a click-iT reaction on the co-cultured cerebellum slices for fluorescent labeling of newly synthesized DNA. Nuclear fluorescence is indicative of active DNA synthesis, which was considerably decreased in cells treated with LDE225 (Figure 5A, Appendix A). EdU incorporation relative to area was also decreased in cells treated with SAG or bFGF alone, whereas the combination of SAG plus bFGF rescued proliferation. We also observed increased abundance and intensity of glial fibrillary acidic protein (GFAP)-positive cell protrusions and infiltrates at the site of the tumor cell mass in SAG and SAG plus bFGF-treated slices; this accumulation of GFAP-positive cells was not observed in the samples treated with BGJ398 or LDE225 (Appendix A). BGJ398 and LDE225 treatment also caused morphological alterations in Purkinje cells reminiscent of axon retraction (Appendix A).

These data indicate that the level of SMO activity determines the invasive behavior of the tumor cells in the cerebellar slice tissue. Inactivation of SMO prevents, whereas its exogenous activation by SAG increases tissue invasion. Co-activation of both SMO and FGFRs results in an intermediate phenotype with reduced invasion (compared to control or SAG or bFGF stimulation) but comparable proliferative activity.

## 3. Discussion

In this study, we investigated the crosstalk between the FGFR and the SHH signaling pathways in cell-based models of MB in the context of cell invasion control. We found that SHH pathway activation with the smoothened agonist SAG triggered sustained expression of the GLI1 transcriptional activator and contributed to tissue invasion ex vivo. Repression of SMO activity by LDE225 only moderately impaired cell growth in vitro but abrogated proliferation and tissue invasion ex vivo. FGFR activation caused collagen and tissue invasion, promoted transcription of *HES1*, and resulted in nuclear translocation of activated ERK. Parallel activation of FGFR and SMO blocked GLI1 expression, markedly reduced nuclear ERK activity, and repressed tissue invasion ex vivo. Neither in vitro nor ex vivo did we observe altered proliferation in response to parallel pathway activation, suggesting that the main impact of the FGFR-SMO crosstalk is on invasion control. In gr3 MB cells, SMO activation did not cause a detectable effect on proliferation and invasion, whereas blockade of either FGFRs or SMO reduced proliferation.

The impact of FGFR activity on MB growth and progression is controversial. FGFR activation was found to block proliferation, to induce differentiation, and to ultimately cause cell death in a cell line derived from a surgical specimen [19]. This activity was later found to be dependent on FGF2 (bFGF) and FGF9 and was only effective in cell lines derived from classic but not desmoplastic tumors [20]. In granule cell precursors (GCPs) isolated from mice, the putative cells of origin of MB, SHH promoted and bFGF repressed proliferation and also ablated GLI1 target gene expression induced by SHH [4]. The antiproliferative effect of bFGF on GCPs was confirmed throughout postnatal development of the GCPs, and both pre-implantation exposure to and injection of bFGF into MB tumors of Ptch+/− mice blocked tumor growth [5]. In contrast to the anti-proliferative effect found in these murine SHH MB models, Zomerman et al. found that bFGF stimulation significantly increased proliferation in human MB cell lines and noted high levels of bFGF release in all lines tested [21]. Our data on the gr3 line HD-MBO3 support this observation, as blockade of FGFR slows proliferation, and bFGF stimulation causes transcription of *MKI67* and *HES1*, two genes involved in proliferation control. They also corroborate our previous findings, where genetic blockade of FGFR signaling impaired tissue invasion ex vivo and pharmacological inhibition of FGFR signaling reduced growth and progression in a gr3 MB in vivo model [3]. Thus, the consequence of FGFR activation on tumor cell growth and progression may depend on the organism’s developmental state and corresponding microenvironmental cues. In support of this is our observation herein, where the activation of SMO in the context of repressed FGFR leads to very robust *GLI1* expression. Conversely, the activation of FGFR causes robust nuclear ERK activation and represses SHH signaling, GLI1 expression, and pro-invasive functions of activated SMO, which may explain the tight spatio-temporal control of FGFR signaling during development, where both SMO and FGFRs play essential roles. GLI1 repression may also explain why engraftment and growth of *Ptch*+/− derived tumors were blocked by bFGF, as this likely caused repression of GLI1 activity in a susceptible phase of tumor growth [5]. Whether this was a consequence of repressed invasion, which we observed in cerebellar slices when both pathways were activated in parallel, remains to be elucidated. Our findings of GLI1 repression by FGFR signaling are in line with [9] and additionally indicate the lack of implication of c-jun N-terminal kinase (JNK), phosphatidylinositol 3’kinase (PI3-K), and protein kinase C (PKC) in FGFR-induced repression of GLI1. This is in contrast to the repression of GLI1 target genes by bFGF in GCPs, where JNK inhibitors were also found to be effective [4]. 

The fact that SMO activation in a condition with complete FGFR inhibition confers moderate invasive properties independent of FGFR indicates that SMO activity toward cell locomotion is constitutively repressed by FGFR signaling. In the absence of activated FGFRs, SMO signaling may thus contribute to tumor cell dissemination and accelerate expansion and spreading of the tumor independent of its growth-promoting functions. Repression of the growth-promoting functions by LDE225 in DAOY cells revealed susceptible and resistant cells in this laboratory line. Proliferation of the gr3 MB line HD-MBO3 is moderately susceptible to LDE225 and requires FGFR signaling for maximal proliferation in vitro. This is consistent with our observation of BGJ398 effect in vivo, where both invasion and tumor growth were blocked with this FGFR inhibitory compound [3].

The small selection of genes analyzed does not provide a clear indication that SMO activation negatively affects FGFR-induced transcription, as the increased expressions of *HES1*, *MK67*, and *MBI1* in bFGF-stimulated cells were not repressed by SAG treatment. Increased *MKI67* transcription in HD-MBO3 cells corroborates the observation that FGFR activation in gr3 MB promotes proliferation and that therapeutic targeting FGFR signaling with small molecule compound inhibitors in this subgroup could provide a clinical benefit. 

Cerebellum tissue invasion of DAOY MB cells is exquisitely sensitive to LDE225. Parallel activation of FGFR and SHH signaling partially phenocopies SMO inhibition by LDE225 and reduces invasion. LDE225 treatment can repress phosphorylation of focal adhesion kinase (FAK) and paxillin [22,23], two molecules critically implicated in integrin-dependent motility and invasiveness in numerous cancers, including MB [24]. SHH signaling promotes epithelial to mesenchymal transition (EMT) and lymph node invasion in bladder cancer [25,26] and hypoxia-induced up-regulation of cancer stem cell genes and EMT in cholangiocarcinoma [27]. Wang et al. [28] furthermore proposed that EMT in esophageal adenocarcinoma is mediated through increased GLI1 expression and is associated with PI3-K AKT signaling. Whether SMO-dependent tissue invasion is mediated through GLI1 in MB is unclear. Our data indicating that FGFR blockade by BGJ398, which triggers *GLI1* expression, is necessary for SAG-induced collagen I invasion, support a mechanism depending on GLI1. However, more direct functions independent of transcriptional control may also need to be considered, in particular as we observed direct repression of nuclear ERK by SAG, which cannot be explained by GLI1 transcriptional activity. The only partial phenotype triggered by FGFR-SMO co-activation indicates transcriptional control of invasion rather than direct repression of—for example—FAK phosphorylation. Despite the remarkable efficacy of LDE225, it is unlikely to be effective in all SHH-driven MB, particularly not in those dependent on oncogenic alterations in the SHH pathway downstream of SMO. Co-treatment with bFGF and SAG does not cause the near-complete blockade of EdU incorporation observed with LDE225, suggesting that the mechanisms promoting invasion and proliferation are not identical. In vitro, blockade of proliferation by LDE225 treatment was only partial with a subset of cells being resistant. The ex vivo analysis did not completely recapitulate this finding, as we observed only a few proliferating cells. However, since the tumor cell mass in LDE225-treated organotypic cerebellum slice cultures (OCSCs) was relatively large at endpoint, it is also possible that either proliferation-arrest occurs in all cells after a lag phase or the low-fetal bovine serum (FBS) medium in the in vitro culture supplied pro-proliferative signals. 

An intriguing observation of our study is that SMO activation causes sustained repression of nuclear accumulation of bFGF-activated ERK. ERK has numerous targets in the nucleus [29], and the level of ERK activation in the nucleus was shown to balance proliferation (if nuclear ERK was high) versus differentiation (if nuclear ERK was low) in stem and progenitor cells [30]. Nuclear translocation of ERK is initiated by phosphorylation of its regulatory Tyr and Thr residues followed by exposure of the nuclear localization signal (NLS), subsequent phosphorylation by CKII, nuclear import through importin 7, and the regulation of up to 125 substrates (reviewed in [29]). Nuclear ERK substrates include transcription factors including c-MYC [31] and chromatin modifying enzymes, which makes it difficult to predict the functional consequence of repressed nuclear ERK activity in FGFR-SMO co-activated MB cells. Correlative gene expression analysis between GLI1 and c-MYC expression indicates negative correlation across all subgroups of MB (*r*-value = −0.236, *p* = 4.03^−11^, source: R2 genomics analysis and visualization platform), suggesting that SMO activation could repress c-MYC in MB. In contrast, GLI1 expression correlates positively with MYCN expression across all subgroups of MB (*r*-value = 0.619, *p* = 8.93^−82^, source: R2 genomics analysis and visualization platform). The mechanistic basis and the functional consequences of the repression of FGFR-induced nuclear ERK activation by activated SMO merit further in-depth investigation, as this regulatory network affects key genes such as c-MYC and MYCN, both relevant for MB tumor initiation and progression of different MB subtypes [2]. 

Activation of MAP/ERK signaling was previously shown through SMO agonist Purmorphamine in fibroblast-like synoviocytes and was found to promote proliferation and motility [32]. We observed no increase in pERK by IB after SMO activation, indicating that ERK regulation downstream of SMO is cell-type dependent, and that the modest increase in SAG-dependent invasion in cells with repressed FGFR is not mediated through ERK. We observed a discrepancy between nuclear ERK activation measured by the ERK relocation sensor and ERK phosphorylation assessed by IB. Although we can only speculate about the underlying cause, a general impact of SAG on the regulation of nuclear import and export can be excluded, as the NLS-mediated nuclear translocation of mCherry-nuc was not affected. However, to mechanistically understand and eventually therapeutically exploit the control of nuclear ERK activation by SHH signaling, further investigation of the regulation of CKII activation and importin7 function by SHH signaling will be necessary. 

## 4. Material and Methods

### 4.1. Reagents

Kinase inhibitors BGJ398 (FGFRs, S2183), Go6983 (PKCs, S2911), LY2157299 (PI3-K, S2230), and SP600125 (JNK, S1460), as well as SMO inhibitor LDE225 (S2151) were purchased from Selleckchem, Houston, TX, USA. bFGF (100-18B), HGF (100-39), EGF (100-47), and SAG (9128694) were purchased from PeproTech EC LtD (London, UK). 

### 4.2. Cells and Cell Culture

DAOY human MB cells were purchased from the American Type Culture Collection (ATCC, Rockville, MD, USA). HD-MBO3 [17] was generously provided by Till Milde (DKFZ, Germany). DAOY cells were cultured as described in [33]. DAOY Lifeact-enhanced green fluorescent protein LA-EGFP cells were produced by lentiviral transduction of DAOY cells with pLenti-LA-EGFP. Cell line authentication and cross-contamination testing were performed by Multiplexion GmbH (Heidelberg, Germany) by single nucleotide polymorphism (SNP) profiling.

### 4.3. Mouse Maintenance

Mouse protocols for organotypic brain slice culture were approved by the Veterinary Office of the Canton Zürich (Approval ZH134/17). Wild type C57BL/6JRj pregnant females were purchased from Janvier Labs (Le Genest-Saint-Isle, France) and were kept in the animal facilities of the University of Zürich Laboratory Animal Center.

### 4.4. qrt-PCR Analysis of Gene Expression

In this study, 2 × 10^5^ DAOY wild type or 2.5 × 10^5^ HDMBO3 wild type cells were seeded per well in 6 well plates in complete growth medium and incubated overnight at 37 °C. Medium was replaced with serum-free medium. After overnight incubation at 37 °C, cells were treated with bFGF (100 ng/mL), SAG (100 nM), BGJ398 (1 µM), or in combination for 48 h with a treatment change after 24 h. For quantitative real-time PCR (qRT-PCR) analysis of target genes, total RNA was isolated using Qiagen RNeasy Mini Kit (74106, Qiagen, Hilden, Germany). Then, 1 μg of total RNA was used as a template for reverse transcription, which was initiated by random hexamer primers. The cDNA synthesis was carried out using High Capacity cDNA Reverse Transcription Kit (4368813, Applied Biosystems, Foster City, CA, USA). qRT-PCR was performed using PowerUp Syber Green (A25776, Thermo Scientific, Waltham, MA, USA) under conditions optimized for the ABI7900HT instrument. The ΔΔCT method was used to calculate the relative gene expression of each gene of interest.

### 4.5. Spheroid Invasion Assay (SIA)

In this experiment, 2000 cells/100 μL per well were seeded in cell-repellent 96 well microplates (650790, Greiner Bio-one). The cells were incubated at 37 °C overnight to form spheroids. Then, 70 µL of the medium were removed from each well, and there remained a medium with spheroid overlaid with a solution containing 2.5% bovine collagen 1. Following the polymerization of collagen, fresh medium was added to the cells and treated with growth factors and/or with inhibitors. The cells were allowed to invade the collagen matrix for 24 h, after which they were fixed with 4% PFA and stained with Hoechst. Images were acquired on an Axio Observer 2 mot plus fluorescence microscope using a 5× objective (Zeiss, Munich, Germany). Cell invasion is determined as the average of the distance invaded by the cells from the center of the spheroid, which was determined using automated cell dissemination counter (aCDc) with our cell dissemination counter software aSDIcs [15]. 

### 4.6. Immunoblot (IB)

To assess treatment effects on target proteins, 2.5 × 10^5^ DAOY wild type or 2.5 × 10^5^ HDMBO3 wild type cells were seeded per well in 6 well plates in complete growth medium and incubated overnight at 37 °C. Medium was replaced with serum-free medium. After overnight incubation at 37 °C, cells were treated with bFGF (100 ng/mL), SAG (100 nM), BGJ398 (1 µM), or in combination for 24–48 h. Treatments were changed every 24 h. Cells were lysed using RIPA buffer and processed for immunoblot (IB) with antibodies against GLI1 (1:1000, Cell Signaling Technologies, Danvers, MA, USA), ERK1/2 (1:1000, Cell Signaling Technologies), and phospho-ERK1/2 (1:1000, Cell Signaling Technologies, Danvers, MA, USA). Loading was normalized using GAPDH (1:1000, Cell Signaling Technologies) or Tubulin (1:1000, Sigma) detected on the same membrane. HRP-linked secondary antibodies (1:5000, Cell Signaling Technologies) were used to detect the primary antibodies. Chemiluminescence detection was performed using ChemiDoc Touch Gel and Western Blot imaging system (BioRad, Hercules, CA, USA) and FujiFilm LAS 3000 (Bucher Biotech, Basel Switzerland) Integrated density of immuno-reactive bands was quantified using ImageJ open source image processing program (https://imagej.net/). 

### 4.7. Cell Fractionation

Cell fractionation assay was performed to determine phosphorylated ERK level in subcellular compartment. The 3 × 10^6^ DAOY cells were seeded in 10 cm dishes 24 h before drug treatment, and cells had fetal bovine serum deprivation overnight (12 h). The next day, DAOY cells were pre-treated or not with LDE-225 (10 µM) and completed after 2 h with bFGF (100 ng/mL) and/or with SAG (100 nM) for 10 or 90 min. Following the recommendations of the provider, we used the FractionPREP^TM^ Cell fractionation kit (K270-50, BioVision, Milpitas, CA, USA) to obtain the cytosolic C, the membrane/particulates M, and the nuclear/cytoskeletal N subcellular protein fractions. During the fractionation, washing steps were done twice between each buffer with cold EDTA-EGTA (1 mM) (Fluka Biochemika, Buchs, Switzerland) containing cocktails of protease and phosphatase inhibitors (Roche). We also added phosphatase inhibitor cocktail in all kit buffers. To perform immunoblots, we added in the final protein fractions, Laemmli buffer (BioRad) and DTT (50 mM) (Sigma). Samples were sonicated and denatured before deposit of 3% of final volume of each fraction on SDS-PAGE gels. We processed for IB with anti-phosphorylated ERK (#9101, Cell Signaling, 1:1000) and anti-total ERK (#9102, Cell Signaling Technologies, 1:1000). To check the correct cellular fractionation, we used anti-FRS2 (Sc-8318, Santa Cruz Biotechnology, Santa Cruz, USA, 1:200) and anti-AIF (Apoptosis Inducible Factor, D39D2, #5318, Cell Signaling Technologies, 1:1000) for the membrane/particulate M fraction, anti-Histone H3 (#4499, Cell Signaling Technologies, 1:2000) and anti-HDAC2 (3F3, #5113, Cell Signaling Technologies, 1:1000) for the nuclear/xytoskeletal N fraction, and anti-GAPDH (#2118, Cell Signaling Technologies, 1:2000) for the xytosolic C fraction. Loading was normalized using GAPDH detected on the same membrane. Relative pERK and ERK protein levels were determined compared to untreated cells. HRP-linked secondary antibodies (Cell Signaling, 1:10.000) were used to detect the primary antibodies. Chemiluminescence detection was performed using ChemiDoc Touch Gel and Western Blot imaging system (BioRad, Hercules, CA, USA) and FujiFilm LAS 3000 (Bucher Biotech AG, Basel, Switzerland. Integrated density of immuno-reactive bands was quantified using ImageJ.

### 4.8. Ex vivo Organotypic Cerebellum Slice Culture (OCSC)

Wild type C57BL/6JRj mice pups were sacrificed at postnatal day (PND) 8–10 by decapitation. Cerebella were dissected and placed in cold Geys balanced salt solution containing kynurenic acid (GBSSK) and then embedded in 2% low melting point agarose gel. Solidified agarose blocks were glued onto the vibratome (VT 1200S, Leica, Wetzlar, Germany) disc with Roti Coll1 glue (0258.1 Carl Roth, Karlsruhe, Germany), mounted in the vibratome chamber filled with cold GBSSK, and 350 μm thick sections were cut. Slices were transferred to petri dishes filled with cold GBSSK. Millipore inserts (PICM 03050, Merck Millipore, Burlington, VT, USA) were placed in six well plates filled with 1 mL cold slice culture medium (SCM) onto which the slices were then transferred using a Rotilabo-embryo spoon (TL85.1, Carl Roth GmbH, Karlsruhe, Germany). A maximum of three slices were placed per insert, and excess medium was removed. Slices were monitored for any signs of apoptosis, and media was changed daily for the first week and once in two days thereafter. Tumor spheroids were formed with DAOY LA-EGFP cells. The co-culture was treated with bFGF (100 ng/mL), SAG (100 nM), LDE225 (10 µM), or BGJ398 (10 µM). Spheroids were incubated for 5 days. Following the treatment, the co-cultures were fixed as described in [18]. After PFA fixation, the slices were incubated in standard cell culture trypsin EDTA and incubated at 37 °C in a humidified incubator for 23 min. The slices were blocked in phosphate buffered saline (PBS) containing 3% fetal calf serum, 3% bovine serum albumin (BSA), and 0.3% triton × 100 for 1 h at room temperature (RT). Primary antibodies were diluted in the blocking solution and incubated overnight on a shaker at 4 °C. Following 3 washes at RT using 5% BSA in PBS, secondary antibodies were incubated for 3 h at RT. The inserts were flat mounted in glycergel mounting medium (C0563, Dako, Jena, Germany). The slice-spheroid co-cultures were stained for GFAP and calbindin, and three-color image acquisition was performed on an SP8 Leica confocal microscope (Leica Microsystems, Mannheim, Germany).

### 4.9. ERK-SKARS

Next, 5000 cells/100 µL were seeded in 96-well plates (Greiner µClear, Greiner Bio One GmbH, Kremsmünster, Austria) in full growth medium. The medium was replaced with serum-free medium after 6 hours and was incubated overnight at 37 °C. After starvation, bFGF and/or SAG were added simultaneously using a multichannel pipette (Gilson, Middleton, WI, USA). Serum-free medium was added to control conditions. The image acquisition was started 1 min after treatment with a Nikon Ti2 widefield microscope using a 20× objective (Nikon Instruments Inc., Melville, NY, USA). The chamber for the plate was constantly held at 37 °C in humidified air (95% air, 5% CO_2_). The cells were imaged for 25–35 ms using the excitation wave lengths of 531 nm (red channel for mCherry-nuc-9) and 482 nm (green channel for SKARS). The intervals between the acquisitions were set to 60 s. The images were analyzed using CellProfiler (version 2.2.0, free open source software, https://cellprofiler.org/).

### 4.10. Cell Proliferation Analysis

To assess cell proliferation, cells were stained with CellTrace Violet cell proliferation kit (C34571, ThermoFisher Scientific) according to the manufacturer’s protocol. Labeled cells were seeded at the concentration of 6 × 10^4^ cells in a 6-well plate with complete growth medium. The medium was then replaced after 6 hours with media containing 1% FBS. After overnight incubation at 37 °C, cells were treated with bFGF (100 ng/mL), SAG (100 nM), BGJ398 (1 µM), LDE225 (10 µM), or in combination for 24, 48, or 72 h. Media containing treatment agents was changed every 24 h. Fluorescence dilution of the dye was measured at the indicated time points by flow cytometry using an LSRFortessa (BD Bioscience, San Jose, CA, USA) and analyzed using FlowJo Software v10.0 (BD Bioscience, San Jose, CA, USA). Treatment with 10 µM of Mitomycin C (M4287, Sigma Aldrich, St Louis, MO, USA) for 16 h was used as a negative control for cell proliferation. For the proliferation curve, 8 × 10^4^ DAOY cells were seeded in a 6-well plate with complete growth medium. The medium was then replaced after 6 h with media containing 1% FBS. After overnight incubation at 37 °C, cells were treated with bFGF (100 ng/mL), SAG (100 nM), BGJ398 (1 µM), LDE225 (10 µM), or in combination. Cells were trypsinized and counted every 24 h over a 72 h period using the Trypan blue exclusion method (Thermo Fisher, Waltham, MA, USA).

### 4.11. Confocal Microscopy

Confocal microscopy of cerebellar slice cultures (Leica Microsystems, Mannheim, Germany) was performed as described in [18].

### 4.12. Gene Expression Analysis

Gene expression data were obtained from the R2 genomics and visualization platform (http://hgserver1.amc.nl/cgi-bin/r2/main.cgi, accessed 12 August 2019) using the Tumor Medulloblastoma-Cavalli-763 dataset [2].

## 5. Conclusions

FGFR activation in MB cells promotes an invasive phenotype in vitro in a 3D collagen invasion model and ex vivo in tissue slices, which is accompanied by increased expression of *HES1*, *MKI67*, and *BMI1*. Concomitant FGFR activation represses SMO-induced expression of Gl1, a key effector of SHH pathway activation. Although such parallel activation of SHH signaling does not impede FGFR-driven expression of *HES1*, *MKI67*, and *BMI1*, it ablates bFGF-induced nuclear translocation and accumulation of activated ERK. The functional consequence of parallel activation of FGFR and SHH signaling ex vivo in tissue slices is the repression of the invasive phenotype induced by FGFR or SHH signaling alone. We conclude from this and from previously published data that, on the one hand, SMO activation promotes proliferation and contributes to tissue invasion through transcriptional control of a set of genes that mediate EMT. On the other hand, FGFR activation promotes invasiveness by directly targeting actin modulators, leading to a transcription-independent pro-invasive phenotype and to proliferation through MAP kinase pathway activation. Concomitant signaling through both pathways ablates GLI1 induction and prevents accumulation of activated ERK in the nucleus, together resulting in a non-invasive cellular state. The susceptibility of the cells to concomitant exposure may depend on the differentiation stage of the cell and the intrinsic status of activation of the SHH pathway with granule cell precursor and early stage MB tumor cells that are highly dependent on SHH signaling for proliferation being more susceptible to FGFR-mediated perturbations. 

## Figures and Tables

**Figure 1 cancers-11-01985-f001:**
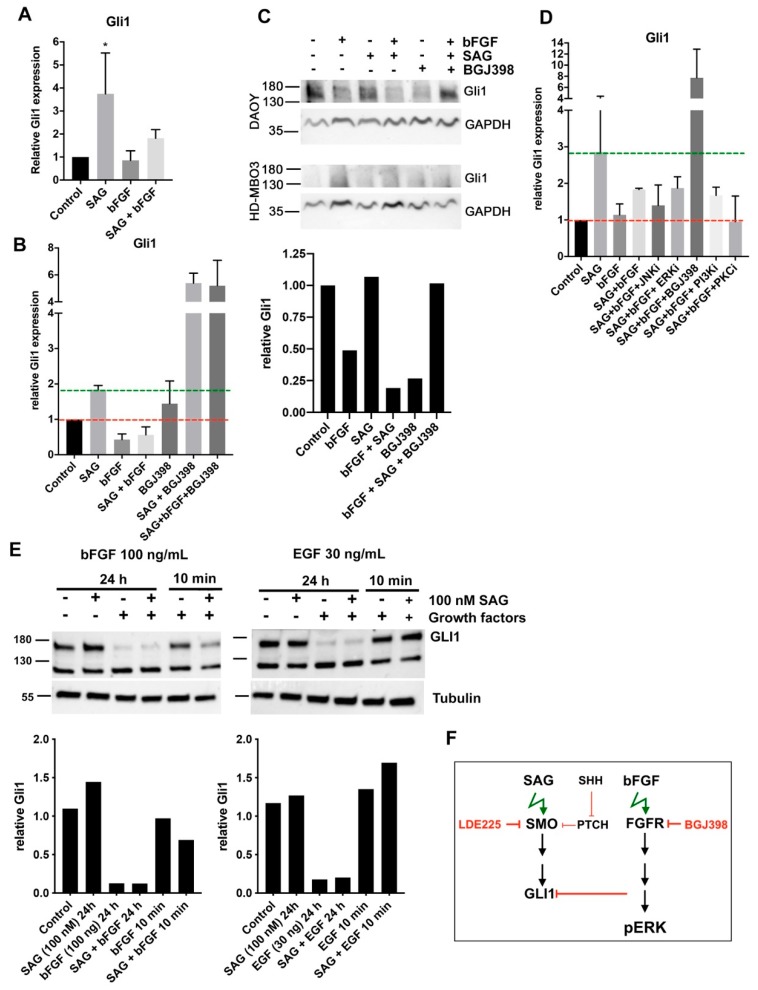
Growth factor signaling represses GLI1 expression. (**A**) qrt-PCR analysis of *GLI1* expression in DAOY cells stimulated with smoothened (SMO) agonist (SAG) (100 nM), basic fibroblast growth factor (bFGF) (100 ng/mL) or in combination for 24 h (*n* = 3, mean and SD, * *p* < 0.05). (**B**) qrt-PCR analysis of BGJ398 (1 µM) effects on SAG and bFGF-induced *GLI1* expression (*n* = 2, mean and SD). (**C**) Immunoblot (IB) analysis of GLI1 expression in response to treatment as in C. No GLI1 expression at protein levels was detected in the gr 3 medulloblastoma (MB) line HD-MBO3. Relative integrated pixel densities of GLI1 bands in DAOY cells are shown below (normalized to Glycerinaldehyd-3-phosphat-Dehydrogenase (GAPDH). (**D**) qrt-PCR analysis of kinase inhibitors against c-jun N-termina kinase (JNK), extracellular-signal regulated kinase (ERK), phosphatidylinositol 3’kinase (PI3K), and protein kinase C (PKCs) (all at 1 µM) effects on SAG plus bFGF-induced *GLI1* expression (*n* = 2, mean and SD). (**E**) Upper: IB analysis of SAG-induced GLI1 expression after 24 h or 10 min stimulation with bFGF (100 ng/mL) or epidermal growth factor (EGF) (30 ng/mL). Right: Integrated densities of GLI1 bands relative to tubulin. (**F**) Schema depicting the observed impact of fibroblast growth factor (FGF)-receptor (FGFR) signaling on GLI1 expression.

**Figure 2 cancers-11-01985-f002:**
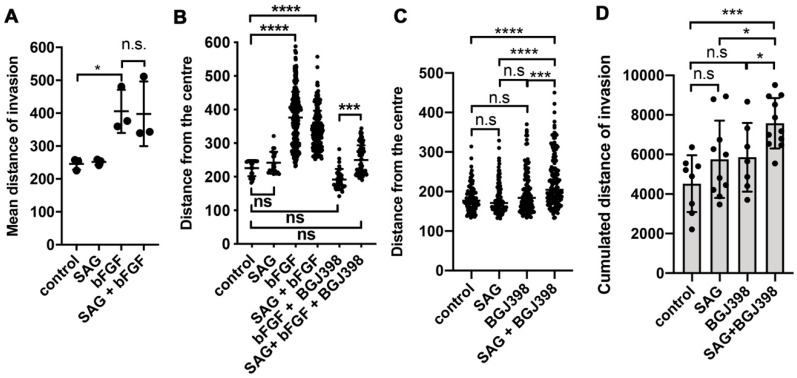
SMO activation does not repress bFGF-induced collagen invasion. (**A**) Mean distance of invasion quantified using Spheroid Invasion Assay (SIA) was compared between treatments as indicated. Each dot represents mean of independent experiment (* *p* = 0.0152, n.s. = not significant unpaired *T*-test). (**B**) Distance of invasion after treatments as indicated. Each dot represents a cell. Data from multiple spheroids are combined (*** *p* < 0.001, **** *p* < 0.0001, one-way ANOVA with Bonferroni’s multiple comparisons test). (**C**) Analysis of BGJ398 impact on distance of invasion compared to BGJ398 plus SAG (*** *p* < 0.001, **** *p* < 0.0001, one-way ANOVA with Bonferroni’s multiple comparisons test). (**D**) As C but total invasion was calculated from the cumulated invasion distances of all cells. Each dot represents the cumulated invasion distance of one spheroid. Mean and SD are shown (* *p* < 0.05, *** *p* < 0.001, n.s. = not significant, unpaired *T*-test).

**Figure 3 cancers-11-01985-f003:**
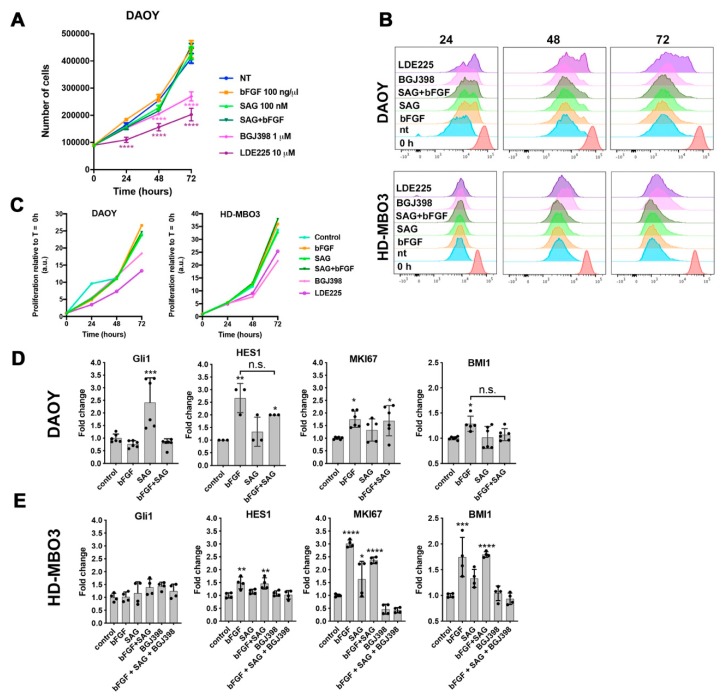
SMO activity is necessary for proliferation without affecting bFGF-dependent gene expression. (**A**) DAOY cells were counted every 24 h over a 72 h period to determine treatment effects on proliferation. Data represent mean values and SEM of three independent experiments. (**B**) CellTrace Violet dye dilution analysis in DAOY and HD-MBO3 cells by flow cytometry every 24 h over a 72 h period with treatments as in A. The fluorescence intensity of the dye measured immediately after labeling the cells at the time of seeding is T0. (**C**) Relative mean fluorescence intensity (MFI) plots of CellTrace Violet dye measurements at indicated times normalized to MFI at T = 0 h (red histogram in B). (**D**,**E**) Gene expression analysis by qrt-PCR after 48 h of the indicated treatments in DAOY (**D**) and HD-MBO3 (**E**) cells (mean and SD, * *p* < 0.05, ** *p* < 0.01, *** *p* <0.001, **** *p* <0.0001, one-way ANOVA with Tukey’s multiple comparisons test).

**Figure 4 cancers-11-01985-f004:**
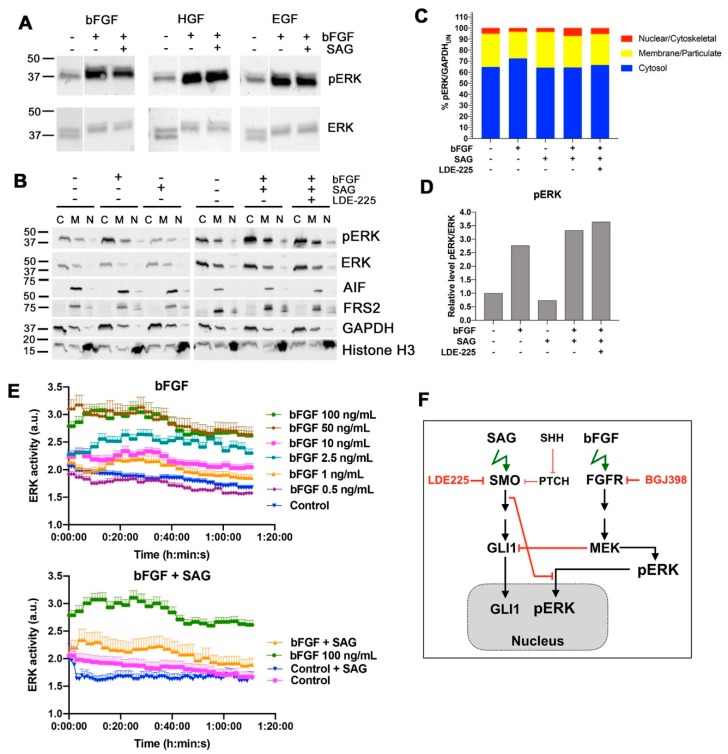
SMO activation represses nuclear ERK activity after bFGF stimulation. (**A**) IB analysis of pERKT202Y204 and ERK after 10 min stimulation with bFGF (100 ng/mL) or hepatocyte growth factor (HGF) (20 ng/mL) or EGF (30 ng/mL). (**B**) IB analysis of proteins indicated in cytosolic (C), membrane/particulate (M), and nuclear/cytoskeleton (N) fractions. (**C)** Integrated densities of pERK bands relative to the control (UN: untreated) GAPDH in the different cellular fractions. (**D**) Relative pERK integrated chemiluminescence intensities of all three fractions in C relative to ERK. (**D**) Cumulated integrated fluorescence intensities of all three fractions in C relative to control. (**E**) Time course of ratio nuclear/cytosolic ERK/SKARS (ERK activity) with indicated treatments. Mean and SEM are shown. (**F**) Schema depicting the proposed impact of SMO activation on nuclear translocation of activated ERK in the SHH-FGFR crosstalk.

**Figure 5 cancers-11-01985-f005:**
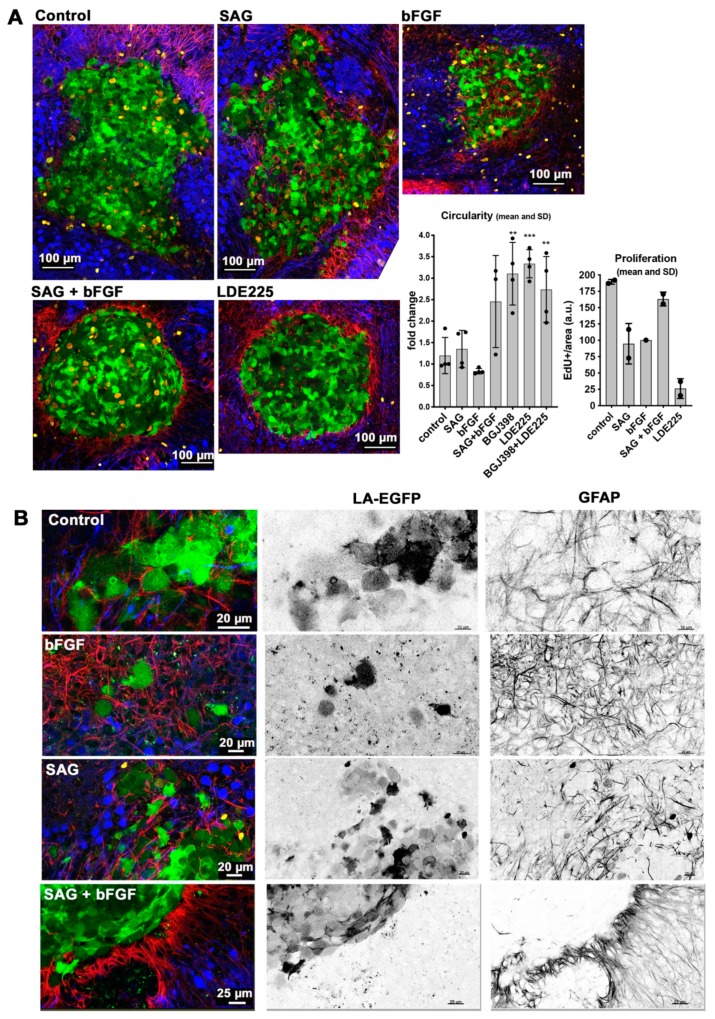
Combined, exogenous activation of SHH and FGFR signaling prevents tissue invasion ex vivo. (**A**) Confocal microscopy analysis of organotypic cerebellum slice cultures (OCSCs) five days after spheroid implantation. DAOY tumor cells express LA-EGFP. Anti-GFAP is in red, anti-Calbindin (Purkinje cells) in blue. Right quantification of circularity (*n* = 4, ** *p* <0.01, *** *p* <0.001, one-way ANOVA with Bonferroni’s multiple comparisons test) and proliferation (*n* = 2). (**B**) Higher-resolution confocal microscopy images of LA-EGFP-positive tumor cells at the spheroid-tissue margins.

**Figure 6 cancers-11-01985-f006:**
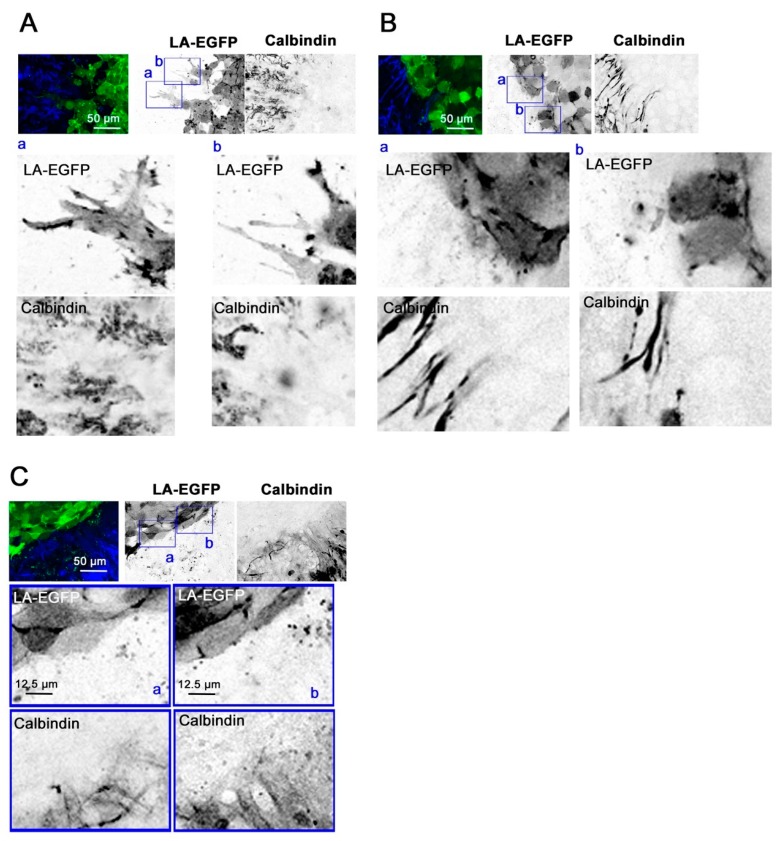
SMO inhibition or bFGF plus SAG co-stimulation restrict invasive phenotype. Confocal microscopy analysis of control (**A**) LDE225 (**B**) and bFGF plus SAG (**C**) co-stimulated OCSCs. Magnifications in (a) and (b) are 4× images of the boxed areas in the upper row. Inverted greyscale was used for better visualization of stains.

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
