# Peer review of "Crosstalk between SHH and FGFR Signaling Pathways Controls Tissue Invasion in Medulloblastoma"

_cancers, 2019, doi:10.3390/cancers11121985_

Round 1

Reviewer 1 Report

The work by Neve et al deals with the FGF-Hedgehog (Hh) crosstalk in medulloblastoma, a Hh-driven tumor of the pediatric cerebellum. The manuscript centers on the seemingly paradoxical activation of two signaling pathways (FGF and Hh), whereby one of them (FGF) inhibits the other (Hh). The study is well performed, well written and clearly structured. It is definitely of interest for people in the field. For instance, an interesting finding is the fact that SMO activation results in suppression of nuclear ERK translocation/activation. Certain points however, should be clarified to support the study:

Major points:

Fig. 1D: No GLI1 induction with SAG. In Fig1B there is clear induction.

Fig. 1E: This is an interesting finding and is reminiscent of data on activated RAS and Hh signaling (Lauth et al., NatStructMolBio, 2010). The authors should test also the combinations of inhibitors of PI3K/JNK/PKC/ERK.

Fig. 1F: Short term treatment (10min with FGF or EGF) apparently induces GLI1 expression. The authors should discuss this topic.

Does FGF affect ciliogenesis in Daoy cells?

Fig. 4A: Western blots are cut in two. Do the bands come from the same membrane? This has to be stated in the legend. Also, the total ERK bands run at different heights. Do they have different molecular weights or is it the result of band cropping and assembly?

Minor points:

Page 2, line 66 (Introduction): The authors speak of myristoylation of GLI proteins. Can the authors provide a reference?

Fig 1B right panel: Is this a mean of several experiments (n=?) or what exactly is it? Please state properly in figure legend.

General nomenclature throughout the article: For example page5, line 164: Human is usually all capital, mouse is only first letter capital. As written by the authors here, Gli1 would be mouse and HES1 would be human.

Fig 4E: Time course: In minutes or something else??

Author Response

Reveiwers comments in italic, bold

Comments and Suggestions for Authors

The work by Neve et al deals with the FGF-Hedgehog (Hh) crosstalk in medulloblastoma, a Hh-driven tumor of the pediatric cerebellum. The manuscript centers on the seemingly paradoxical activation of two signaling pathways (FGF and Hh), whereby one of them (FGF) inhibits the other (Hh). The study is well performed, well written and clearly structured. It is definitely of interest for people in the field. For instance, an interesting finding is the fact that SMO activation results in suppression of nuclear ERK translocation/activation. Certain points however, should be clarified to support the study:

Major points:

Fig. 1D: No GLI1 induction with SAG. In Fig1B there is clear induction.

We see a clear induction at mRNA level (Figs 1 A,B,D) but not at protein level (Fig, 1C). This was stated in the manuscript line 108: “These findings show that the activation of SMO promotes GLI1 transcription in DAOY cells without increasing GLI1 protein.» The quantification of the western blot in Fig. 1C is next to Fig. 1B. Maybe reviewer 1 was referring to that panel?

Fig. 1E: This is an interesting finding and is reminiscent of data on activated RAS and Hh signaling (Lauth et al., NatStructMolBio, 2010). The authors should test also the combinations of inhibitors of PI3K/JNK/PKC/ERK.

This indeed would be interesting, in particular if a difference would be noted. However, to reveal synergistic or complementary activities of kinase-mediated GLI1 repression would be outside of the scope of the present manuscript.

Fig. 1F: Short term treatment (10min with FGF or EGF) apparently induces GLI1 expression. The authors should discuss this topic.

Neither bFGF nor EGF stimulation does increase Gli1 protein compared to untreated control. However, we observed decreased Gli1 protein after bFGF and SAG co-stimulation. This is likely due to GLI1 phosphorylation by activated MEK (Lu et al. Oncogene 2018), which leads to its proteolytic degradation.  

Does FGF affect ciliogenesis in Daoy cells?

We have shown that bFGF triggers filopodia formation in DAOY cells (Santhana Kumar et al. Cell Reports 2018). However, we have not tested primary cilium formation or maintenance in response to FGFR activation or repression. It is indeed possible that the profound effect of FGFR signaling on cortical actin dynamics affected primary cilium function and thus SHH signaling.

Fig. 4A: Western blots are cut in two. Do the bands come from the same membrane? This has to be stated in the legend. Also, the total ERK bands run at different heights. Do they have different molecular weights or is it the result of band cropping and assembly?

Yes, all bands are from the same membrane. These and all other membranes are now shown at full size in figures S5, S6 and S7. In Fig. 4, membranes were cut due to additional treatments not relevant for this study.

Size difference in total ERK: We always observe a mobility shift in total ERK in response to growth factor stimulation (bFGF, EGF or HGF). We interpret this shift as a sign of phosphorylation. Indeed, we usually only observe the shift in stimulated cells and we find the shift to correlate with the intensity of pERK staining.

Minor points:

Page 2, line 66 (Introduction): The authors speak of myristoylation of GLI proteins. Can the authors provide a reference?

This sentence was not correct, it meant to refer to myristoylated regulators (now removed) and acetylation, ubiquitination or sumyolation of GLI1 proteins. We have changed the text accordingly.

Fig 1B right panel: Is this a mean of several experiments (n=?) or what exactly is it? Please state properly in figure legend.

There is no Fig. 1B right. The quantification shown is part of Fig. 1C and depicts the relative intensities of the GLI1 bands shown in Fig. 1C upper. This is described in the figure legend of Fig. 1.

General nomenclature throughout the article: For example page5, line 164: Human is usually all capital, mouse is only first letter capital. As written by the authors here, Gli1 would be mouse and HES1 would be human.

We are grateful to the reviewer to point this out. We have adjusted the nomenclature throughout the article.

Fig 4E: Time course: In minutes or something else??

The time course is in h:min:s. This information was missing in the figure and is now indicated in the X-axis legend of figure 4E.

Reviewer 2 Report

Figure 2B: The authors state (line 133) that "Smo activation in the absence of FGFR signaling can moderately increase the invasion capability of the cells in vitro". However, that doesn't seem true when cells are treated with SAG alone? Can the authors perform an experiment where cells are treated with SAG and BGJ398 to negate off-target effects of the drug?

Overall, a Gli1 KO cell line would be a good way to confirm SAG-induced invasion requires Gli1 expression triggered by FGFR inhibition.

Author Response

Figure 2B: The authors state (line 133) that "Smo activation in the absence of FGFR signaling can moderately increase the invasion capability of the cells in vitro".

However, that doesn't seem true when cells are treated with SAG alone?

SAG alone does not cause a significant increase in invasion. We think this is due to the inhibitory effect of basal FGFR signaling in the cells. Only when FGFR signaling is repressed, i.e. by blocking FGF receptor kinase activity with BGJ398, this effect of Smo activity becomes evident (see below, next point).

 Can the authors perform an experiment where cells are treated with SAG and BGJ398 to negate off-target effects of the drug?

Off target effects of drugs are indeed difficult to assess. To determine the impact of BGJ398 in comparison to SAG, we have performed an additional experiment comparing the effect of SAG alone, BGJ398 alone and the combination of SAG plus BGJ398, now shown in new Fig. 1C and 1D. This experiment showed that neither SAG alone nor BGJ398 alone caused a significant increase in invasion. This excludes to some extent an off-target effect of BGJ398. Only the combination of SAG plus BGJ398 causes significant invasion compared to untreated control, but also compared to SAG alone or BGJ398 alone.

We inserted the text starting line 124 as follows: “Co-treatment with bFGF, SAG and BGJ398 caused a significant increase in collagen I invasion compared to bFGF plus BGJ398 treatment alone. Although BGJ398 only treatment caused some increase in invasion, its impact on invasion is only significant on the distance of invasion (Fig. 1C) as well as on the cumulated distance of invasion (Fig. 1D) when SAG is also present. This indicates that SMO activation can moderately increase the invasion capabilities of the cells in vitro when FGFR signaling is repressed.”

Overall, a Gli1 KO cell line would be a good way to confirm SAG-induced invasion requires Gli1 expression triggered by FGFR inhibition.

Indeed, we are considering this for a more extended study on SMO control of invasion. The number of experiments and controls such a study requests is beyond the scope of the current manuscript.

Reviewer 3 Report

Significant advances have been accomplished over the last years in molecular characterization of medulloblastoma with four major molecular subgroups.

Molecular mechanistic needs additional efforts to be fully elucidated. Therefore the paper of Neve et al. investigating the crosstalk between Shh and FGFR signaling in medulloblastoma is of interest.

However, major points need to be addressed by the authors to increase the impact of the paper

A figure summarizing the signaling pathways findings (the cross talk, activation, inhibition) should be added to the manuscript avoiding the reader to draw it himself along the paragraphs and the paper The data reported by the authors highlight the major role of the microenvironment in cancer oncogenesis. However, the studies were performed exclusively in vitro and ex vivo experiments. In vivo experiments would be very important to validate the results obtained by the authors. Ex vivo recapitulates partly in vivo conditions. Balance/ratio between Smo and FGFR should be specified quantatively Impacts on tumor cell differentiation (IHC markers) have not been explored in addition to invasion and proliferation. This might be of interest in the setting of medulloblastoma

Author Response

Significant advances have been accomplished over the last years in molecular characterization of medulloblastoma with four major molecular subgroups.

Molecular mechanistic needs additional efforts to be fully elucidated. Therefore the paper of Neve et al. investigating the crosstalk between Shh and FGFR signaling in medulloblastoma is of interest.

However, major points need to be addressed by the authors to increase the impact of the paper.

A figure summarizing the signaling pathways findings (the cross talk, activation, inhibition) should be added to the manuscript avoiding the reader to draw it himself along the paragraphs and the paper

A schema depicting a summary of the findings is now included in Figs 1F and 4F. The graphical abstract provides an additional overview.

The data reported by the authors highlight the major role of the microenvironment in cancer oncogenesis. However, the studies were performed exclusively in vitro and ex vivo experiments. In vivo experiments would be very important to validate the results obtained by the authors. Ex vivo recapitulates partly in vivo conditions. Balance/ratio between Smo and FGFR should be specified quantatively Impacts on tumor cell differentiation (IHC markers) have not been explored in addition to invasion and proliferation. This might be of interest in the setting of medulloblastoma

These are excellent suggestions and we are actively pursuing them. However, to exactly quantitate the outcome of a precisely balanced exposure to the two factors is very challenging as it requires for each experimental procedure the titration of both factors. Together with the IHC analyses proposed, I see this as a separate study to broaden the significance of our findings.

Round 2

Reviewer 2 Report

All concerns have been addressed.

Please note that the response "To determine the impact of BGJ398 in comparison to SAG, we have performed an additional experiment comparing the effect of SAG alone, BGJ398 alone and the combination of SAG plus BGJ398, now shown in new Fig. 1C and 1D."- should read as figure 2C and 2D. This is in accordance with the text.

Reviewer 3 Report

The authors have fairly adressed the comments.